# Suicide during pregnancy as a major contributor to maternal suicide among female sex workers in eight low- and middle-income countries: A community knowledge approach investigation

## Research Article

suicide; female sex workers; maternal mortality; prenatal; mental health

**Corresponding author:**
Wendy L. Macias-Konstantopoulos;
Email: wmacias@mgh.harvard.edu

Wendy L. Macias-Konstantopoulos[1,2] , Brian Willis[2], Swarna Weerasinghe[2,3], Emily Perttu[2] and Ian M. Bennett[4]

[1]Center for Social Justice and Health Equity, Department of Emergency Medicine, Massachusetts General Hospital, Harvard School of Medicine, Boston, MA, USA; [2]Global Health Promise, Portland, OR, USA; [3]Department of Community Health and Epidemiology, Faculty of Medicine, Dalhousie University, Halifax, Nova Scotia, Canada and [4]Departments of Family Medicine, Psychiatry and Behavioral Sciences, and Global Health, University of Washington, Seattle, WA, USA

## Abstract

Studies indicate a high burden of mental health disorders among female sex workers (FSWs) in low- and middle-income countries (LMICs). Despite available data on suicidal ideation and suicide attempts among FSWs, little is known about suicide deaths in this hard-to-reach population. This study aims to examine the extent to which suicide is a cause of maternal mortality among FSWs, the contexts in which suicides occur, and the methods used. From January to October 2019, the Community Knowledge Approach method for identifying cause-specific deaths in communities was employed across eight LMICs (Angola, Brazil, the Democratic Republic of the Congo (DRC), India, Indonesia, Kenya, Nigeria, and South Africa). A total of one thousand two hundred eighty FSWs provided detailed reports on two thousand one hundred twelve FSW deaths in the preceding 5 years, including 288 (13.6%) suicides, 178 (61.8%) of which were maternal. Of these maternal suicides, 57.9% occurred during pregnancy (antepartum), 20.2% within two months of delivery (puerperium), and 21.9% in the 2–12 months following delivery (postpartum). The highest proportion of suicides occurred in Nigeria, Kenya, and DRC in sub-Saharan Africa. A total of 504 children lost their mothers to suicide. Further research is needed to identify interventions for suicide risk among FSW mothers.

## Impact statement

Globally, the sex worker community remains under-studied despite significant stigma and elevated risk for poor health outcomes. Research indicates the prevalence of perinatal depression is significantly higher in women from low- and middle-income countries (LMICs) compared to perinatal women from high-income countries (HICs) and in women from vulnerable groups compared to the general population of women. Among the highly stigmatized female sex worker (FSW) population, the estimated prevalence of depression is between 50% and 88%, and approximately 39% are considered at risk for suicide. Risk factors associated with mental health problems among FSWs include substance use, HIV infection, and experiences of violence. A meta-analysis of FSWs in LMICs estimated that the pooled prevalence of past-year and lifetime suicidal ideation was 22.8% and 25%, respectively. Data on suicide attempt rates among FSWs are wide-ranging (2.6% to 44.2%), and even less is known about suicide as a cause of death among FSWs. Using the community knowledge approach method for identifying cause-specific deaths in communities of women, the current study provides new insights into suicide as a cause of mortality among FSWs in LMICs. Approximately 14 % of reported FSW deaths were due to suicide. Nearly two-thirds of all FSW suicides reported were maternal, and the majority of these suicides occurred during the prenatal period. Thus, although the evidence for a protective effect has been inconclusive, any relative protective effect of pregnancy against suicide seen among other perinatal populations in both HICs and LMICs is not observed among the FSW populations in this study. The prenatal period appears to be a higher risk period for suicide in FSWs from the LMICs included in this study, a finding that merits further investigation and is key to identifying potential interventions for reducing maternal mortality due to suicide in this community.

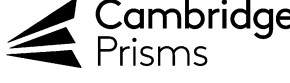



## Introduction

The World Health Organization (WHO) estimates that 4.4% of the world's population (322 million persons) suffers from depression, with more than 80% of the disease burden

concentrated in LMICs (WHO, 2017). Globally, approximately 12% of women experience depression during pregnancy and the 12 months after delivery (Woody et al., 2017).

Compared to perinatal women in HICs, perinatal depression (PND) is significantly higher in women from LMICs (Woody et al., 2017; Roddy Mitchell et al., 2023). PND disproportionally impacts women with social vulnerabilities—such as economic deprivation, intimate partner/gender-based violence, poor social support, HIV infection, migrant status, COVID-19 pandemic stressors—particularly in LMICs (Ajinkya et al., 2013; Fuhr et al., 2014; Fellmeth et al., 2017; Mikšić et al., 2018; Wang et al., 2021; Orsolini et al., 2022; Orsolini et al., 2023; Al-abri et al., 2023). Significantly, PND is not only associated with obstetric and neonatal complications such as premature labor, low birth weight, and preeclampsia (Jarde et al., 2016), but it is also widely recognized as a major contributor to suicide (WHO, 2017).

Worldwide, more than 700,000 people die by suicide each year, and over 77% of these deaths occur in LMICs, according to the WHO (WHO, 2021a). The rate of perinatal suicide, occurring during pregnancy and the 12 months after delivery, ranges from 1.27 to 3.7 per 100,000 live births worldwide (Rodriguez-Cabezas and Clark, 2018). Death by suicide is increasingly recognized as a leading cause of maternal mortality, accounting for about 20% of deaths in the year post-delivery (Lindahl et al., 2005; Chin et al., 2022). Among young Bangladeshi women and adolescents reporting past-year suicide attempts, 88.5% occurred during the postpartum period (Li et al., 2021). Perinatal suicide is associated with preexisting or current psychiatric diagnosis, undesired pregnancy, restricted abortion access, and postpartum depression (Say et al., 2014; Orsolini et al., 2016).

The 12-month postpartum period following delivery has long been associated with higher levels of mood disorder, bipolar disorder, substance use disorder, psychotic disorder, and suicide attempts as compared to during pregnancy (Viguera et al., 2011; Mota et al., 2019). A relative protective effect of the time from conception to birth is, however, inconsistently seen and studies suggest that suicide during pregnancy may be more common than previously recognized (Mauri et al., 2012; Orsolini et al., 2016; Vawda, 2018; Chin et al., 2022; Xiao et al., 2022). A recent systematic review including nearly 6.5 million women found that the global prevalence of suicide attempts was 680 per 100,000 during the antepartum period, three times the prevalence of the 12-month postpartum period (210 per 100,000) (Rao et al., 2021).

Many FSWs experience physical and psychological trauma associated with their life and work circumstances, resulting in a high burden of mental illness among FSWs, including anxiety, depression, post-traumatic stress disorder, and suicidality (Beattie et al., 2020; Millan-Alanis et al., 2021). FSWs in LMICs may experience worse mental health outcomes than other female populations due to the unique psychosocial and financial stressors associated with sex work, including high-risk sexual behaviors, high rates of HIV, high levels of societal stigma, criminalization, poor access to basic health care, discriminatory health services, worse overall health outcomes, client-perpetrated physical, sexual, and psychological violence (e.g., witnessed abuse of children), family rejection, lack of social supports, unfavorable living conditions, harmful substance use, and severe economic marginalization with difficulty meeting basic needs (e.g., food insecurity) for themselves and their children (Ngugi et al., 2012; Muldoon et al., 2015; Wanyenze et al., 2017; Beattie et al., 2020; Millan-Alanis et al., 2021; Panneh et al., 2022; Argento et al., 2021; Martin-Romo et al., 2023).

In a large meta-analysis, the pooled prevalence of depression, suicidal ideation, and suicide attempts among FSWs in LMICs was

several times higher than in the general population (Beattie et al., 2020). The estimated prevalence of depression is between 50% and 88% among FSWs, and approximately 39% are considered at risk for suicide (Martin-Romo et al., 2023). Among FSWs in LMICs, the pooled prevalence of past-year suicidal ideation is estimated to be 22.8%, and the lifetime suicidal ideation prevalence was 25% (Beattie et al., 2020). Substance use, high-risk sexual behaviors, HIV infection, and experiences of violence were specifically associated with suicidal ideation and suicidal behavior (Shahmanesh et al., 2009; Beattie et al., 2020; Martin-Romo et al., 2023). In one study of FSWs in Kazakhstan with a history of drug use, 52.5% reported suicidal ideation in the past week (Velez-Grau et al., 2021).

Data on suicide attempt prevalence among FSW cohorts are wide-ranging from 2.6% to as high as 44.2% (Shahmanesh et al., 2009; Teixeira and Oliveira, 2017; Beksinska et al., 2021), though a recent large meta-analysis estimated the pooled prevalence of suicide attempts among FSWs to be 20% (Millan-Alanis et al., 2021). In a cohort of 1,083 FSWs in South Korea, suicide attempts were independently associated with poor self-rated health, sexually transmitted infections, near-daily alcohol use, and the inability to regularly use condom protection (Jung, 2013).

In contrast to suicide attempts, little is known about death by suicide among FSWs and the extent to which suicide contributes to overall FSW mortality. A recent WHO assessment on the robustness of health information systems across 133 countries concluded that approximately 40% of global deaths remain unregistered and uncounted, with more pronounced data gaps in LMICs (WHO, 2021b). Weak national statistics and civil registries necessitate an alternative scientific method to avoid underestimating mortality data, especially among marginalized, low-resourced populations with poor access to health facilities (Whittaker et al., 2021). Various methods for closing the information gap are leveraged in global health research, including household surveys, capture-recapture analysis of informant-generated lists of deaths, and verbal autopsies (UN, 2005; Roberts et al., 2010; Seal et al., 2021; Bailo et al., 2022).

The community knowledge approach (CKA) is a validated method for accurately identifying cause-specific deaths among members of a defined community with high sensitivity as demonstrated in side-by-side comparison with the household survey method (Qomariyah et al., 2010; Mir et al., 2015; Paul et al., 2018; Gupta et al., 2021). Through the CKA, our study leverages the collective knowledge and memory of FSWs living in peer communities to identify all causes of death among this hard-to-reach vulnerable population of women and their children. As FSWs frequently migrate away from their families of origin or do not disclose their employment as sex workers, the CKA overcomes the challenge of relying on household surveys to collect details on deaths about which families may have little to no information.

The current exploratory study uses the CKA method to examine suicide as a cause of maternal and non-maternal FSW mortality, to understand the contexts in which suicides occur, and to identify the suicide methods used in eight LMICs—Angola, Brazil, DRC, India, Indonesia, Kenya, Nigeria, and South Africa—where women are estimated to account for approximately 20% to 40% of all suicides (WHO, 2021a).

## Methods

### Study design and setting

Death data were collected from January 16 to October 1, 2019 as part of a multicountry cross-sectional cohort study exploring the

causes of mortality among FSWs and their children across eight LMICs (Willis et al., 2022). Given the recognized shortcomings of global health data systems, particularly for stigmatized causes of death and populations (WHO, 2021b), the study was specifically designed to capture data on causes of death among this marginalized, difficult-to-study population. Causes of death among FSWs within primarily urban sex worker social networks were identified using a modified CKA method to leverage the collective knowledge of the FSW community (Paul et al., 2018; Gupta et al., 2021). Country selection criteria included: (1) large FSW population, (2) high maternal mortality rate, (3) high HIV infection rate among FSWs, (4) local partnerships with sex worker organizations (SWOs) and non-governmental organizations (NGOs) providing services to FSWs, and (5) contribution to the geographic diversity of the study. As all five of these conditions needed to be adequately satisfied to the extent possible using data from decentralized and incomplete sources of data (e.g., WHO, UNAIDS, UNFPA), fixed thresholds for the above selection criteria were not enforced. Rather, the available data from a wide range of LMICs were compared to determine the best relative parameters. Considerations beyond the selection criteria included the presence of civil or political unrest, military conflict or humanitarian disaster zones, infectious disease outbreaks, and other such safety travel advisories, the reliability and level of engagement of local partners, and cost and time considerations. Once country study sites were determined, cities within each country were chosen based on in-country transportation logistics, safety considerations, and the established presence of trusted local partners within FSW communities.

## Participants and recruitment

Communities of FSWs were sampled based on the presence and engagement of local partners with FSW members of the community. Potential FSW participants were identified at local bars, brothels, and other common areas of work. To gauge study participation interest, FSWs were informed about the study, its aims, and their role as peer informants on the health of FSWs and their children, including details about deaths occurring within their peer communities. Interested FSWs were screened for study eligibility. Criteria for participation included: (1) age ≥ 18 years, (2) mother of at least one child aged ≤10 years, (3) engaged in full-time sex work for at least 3 years, and (4) socially interactive with other mothers and non-mothers within the FSW community. Although a subset of participants may represent a convenience sample, there was no requirement that study participants receive services from local partners, and word-of-mouth within FSW peer communities facilitated snowball sampling recruitment.

Enrollment for participation included a verbal informed consent process through which participants were notified that participation was voluntary and that withdrawal was possible at any point in time. No information was available to the researchers about those who declined to participate. No personal identifying information was collected or recorded, and participants symbolically noted their consent with either an "x" or check mark on the informed consent form. Recruited FSW attended a single information-gathering group session held in private and safe locations. Study staff collaborated closely with local partners to create a safe space where participants could share information about the deaths of peers. Participants were encouraged to disengage, without fear of repercussion, if at any point in time, they felt triggered or upset by their participation, or if their psychological or emotional well-being felt compromised. Participants were informed they could follow up with the local SWO or NGO partner for assistance with mental health services if they experienced any distress following the information session. Remuneration for their time included food and petty cash ranging from $10 to $14 US dollars (for reference, approximate cost of a meal at a mid-range restaurant) as recommended by local partners.

## Data collection and quality assurance

Information-gathering group sessions were conducted across 24 cities in eight countries. On average, 4–10 study participants attended a group session, lasting approximately 60 min each. Data collection was conducted by the lead investigator (BW) in conjunction with protocol-trained local partner staff. When language translation was required, local partners or partner-approved language interpreters were employed.

Group sessions covered topics related to the health of FSWs and their children, including detailed reports of deaths that occurred in the preceding 5 years (from 2014 to the session date). For reports of death, participants were asked a series of questions about the deceased woman, including her name, date of death, age at the time of death, pregnancy status, gestation age at the time of death, if pregnant, city of death, cause of death, and number of children left behind. In cases of suicide deaths, the method and location of suicide, and the number of children killed in filicide-suicides were collected. As death reports can often involve some component of "storytelling," extensive hand-written field notes were taken to capture details of each death reported beyond the answers to questions posed by the interview guide. The lead investigator and local partners reviewed all death data collected to identify potential duplicates. If any deaths matched the reported details, only the first death was recorded. Data on suicides were entered into a secure electronic database and reviewed by three researchers for accuracy, completeness, and duplication using the original records.

## Data classification and analysis

Suicides were organized by country, age group, and type (non-maternal versus maternal). Maternal suicides were further classified by the perinatal period in which they occurred—antepartum, puerperium, and postpartum—in accordance with the WHO International Classification of Diseases Maternal Mortality (ICD-MM; WHO, 2012). The ICD-MM defines antepartum as corresponding to the pregnant state. The puerperium is defined as the period "within 42 days of termination of pregnancy," while postpartum is the period "more than 42 days but less than 1 year after termination of pregnancy." Given this level of preciseness could not be reasonably expected from peer sources of death reports, the time threshold between the puerperium and postpartum periods was set at 2 months, rather than 42 days. Thus, for the purpose of this study, puerperium was defined as the period "within 2 months of delivery" and postpartum as the "2 to 12 months after delivery."

Methods of suicide categories were defined by investigators (BW, WMK). Two research team members (SW, WMK) independently coded all deaths by the method of suicide employed. Data tables displaying the number of deaths by suicide method and by non-maternal vs. perinatal period were created and shared with all team members for review and adjudication. Discrepancies were resolved through discussion and review of raw data (i.e., field notes, database) and ICD-MM definitions until a consensus was reached. Interrater reliability was assessed by calculating the Cohen's kappa statistic.

Descriptive data analyses were used to present suicide death data. Quantitative data were analyzed using frequencies and percentages across the country, maternal status, and method of suicide. Due to the non-random nature of the sample, no statistical inferences were performed. Using RapidMiner Studio 9.1 software package text mining tools (Mierswa and Klinkenberg, 2018), narrative text data collected from participants sharing detailed reports of deaths in their community, including direct quotes, were analyzed for themes and insights into the circumstances surrounding FSW suicides.

### Ethics

The study protocol, consent forms, and questionnaire were reviewed and approved by the Portland State University Institutional Review Board, USA (protocol #184888). Study protocol and materials were additionally reviewed by all local partners across the eight study countries to ensure compliance with local standards prior to approval and agreement to assist with study conduction.

### Results

One hundred sixty-five information-gathering group sessions were convened across study sites, one thousand two hundred eighty FSWs participated in a group session, and collectively, two thousand one hundred twelve FSW deaths, representing a mean of 1.65 deaths reported per participant across the study. Of the total deaths reported, 288 (13.6%) were attributable to suicide, and the majority of these (*n* = 178, 61.8%) were perinatal suicides occurring during pregnancy or during the puerperium or postpartum periods. The largest proportion of suicide deaths were reported in the sub-Saharan African countries of Kenya, Nigeria, and the DRC, accounting for 242 (84%) of suicide deaths reported (Table 1). The four topics that emerged from the narrative text data analytics were methods of suicide, occurrence of filicide-suicide, determinants of suicide, and caring for very young children.

### Maternal suicide deaths by country and perinatal period

Of the 178 maternal suicides, 103 (57.9%) occurred during the antepartum period during pregnancy, 36 (20.2%) during the puerperium (delivery to two months postpartum), and 39 (21.9%) during the postpartum period (two to twelve months after delivery). The three sub-Saharan African countries of Kenya, Nigeria, and the DRC accounted for 156 (87.6%) of all maternal suicides. Data on maternal suicide by country and perinatal period are presented in Table 2.

### Suicide deaths by age group

The distribution of suicide deaths by age group is provided in Table 3. Among the youngest age group (10–14 years), the one death reported was by suicide. The largest proportion of suicides (*n* = 187, 65.0%) occurred among FSWs aged 20–29 years, for whom suicide accounted for 15.5% of all deaths from all causes. Suicide represented 12.1% and 11% of deaths from all causes among 30–39-year and 15–19-year, respectively. Among FSWs ages ≥40, most suicides were non-maternal, while a predominance of suicides among younger FSWs (<40 years) were maternal. Across age groups, maternal suicide was most common during the antepartum period.

### Methods of suicide

The most reported methods of suicide were poisoning (*n* = 82, 28.5%), hanging (*n* = 67, 23.3%), and drug overdose (*n* = 47, 16.3%). Other methods of suicide included self-inflicted stabbing (or cutting of the wrist/neck), self-immolation, drowning, jumping from a height, discontinuing HIV antiretroviral therapy, and train strikes on railway tracks (Table 4).

### Filicide-suicide

Reports of filicide involved 14 of the 288 total suicides (4.9%) and resulted in the death of 19 children. Up to three children died per filicide-suicide. Reasons for filicide-suicide included family rejection (*n* = 1, 7.1%), lack of resources to care for their children (*n* = 1, 7.1%), and new HIV diagnosis (*n* = 4, 28.6%). Poisoning (*n* = 9, 64.3%) was the most reported method of filicide-suicide, followed by hanging (*n* = 2, 14.3%) and drowning (*n* = 1, 7.1%). The youngest child killed was 2 months old.

**Table 1.** Study countries and suicide deaths

| Country | No. sessions (*N* = 165) | No. session participants (*N* = 1,280) | No. (%) suicides (*N* = 288) | Non-maternal suicides *n* (%) | Maternal suicides *n* (%) | No. (%) female suicides (WHO, 2021a) | Selling and/or soliciting sex were criminalized in 2019* |
|---|---|---|---|---|---|---|---|
| Angola | 12 | 71 | 0 | 0 | 0 | 369 (19.1) | Yes (sell) |
| Brazil | 18 | 80 | 1 (2.6) | 1 (100) | 0 | 3,249 (22.3) | Yes (solicit) |
| DRC | 27 | 270 | 80 (13.2) | 29 (36.2) | 51 (63.8) | 1,206 (20.8) | Yes (solicit) |
| India | 20 | 152 | 26 (19.7) | 15 (57.7) | 11 (42.3) | 72,935 (42.1) | No (if done privately) |
| Indonesia | 12 | 76 | 7 (12.1) | 3 (42.9) | 4 (57.1) | 1,448 (22.1) | Yes (sell) |
| Kenya | 18 | 175 | 90 (17) | 38 (42.2) | 52 (57.8) | 843 (26.2) | Yes (solicit) |
| Nigeria | 32 | 312 | 72 (13.7) | 19 (26.4) | 53 (73.6) | 1,909 (27.2) | Yes (solicit) |
| South Africa | 26 | 144 | 12 (7.1) | 5 (41.7) | 7 (58.3) | 2,913 (21.1) | Yes (sell & solicit) |
| Total | | | | 110 (38.2) | 178 (61.8) | | |

*Global Network of Sex Work Projects, updated through Dec 2021: https://www.nswp.org/sex-work-laws-map/country-list.

**Table 2.** Maternal suicide deaths by country and perinatal period

| Country | No. (%) maternal suicides (N = 178) | No. antepartum suicides n (%) | No. puerperium suicides n (%) | No. postpartum suicides n (%) |
|---|---|---|---|---|
| Angola | 0 | 0 | 0 | 0 |
| Brazil | 0 | 0 | 0 | 0 |
| DRC | 51 (28.7) | 41 (80.4) | 4 (7.8) | 6 (11.8) |
| India | 11 (6.2) | 9 (81.8) | 2 (18.2) | 0 |
| Indonesia | 4 (2.2) | 3 (75) | 0 | 1 (25) |
| Kenya | 52 (29.2) | 23 (44.2) | 15 (28.8) | 14 (27) |
| Nigeria | 53 (29.8) | 24 (45.3) | 13 (24.5) | 16 (30.2) |
| South Africa | 7 (3.9) | 3 (42.8) | 2 (28.6) | 2 (28.6) |
| Total | | 103 (57.9) | 36 (20.2) | 39 (21.9) |

**Table 3.** Suicide deaths by age group

| Age group (N = deaths from all causes) | No. (%) suicides (N = 288) | No. (%) suicides | | | | Suicide as a % of deaths from all causes |
|---|---|---|---|---|---|---|
| | | Non-maternal | Antepartum | Puerperium | Postpartum | |
| 10–14 years (N = 1) | 1 (0.3) | 0 | 1 (100) | 0 | 0 | 100% |
| 15–19 years (N = 282) | 31 (10.8) | 8 (25.8) | 18 (58.1) | 3 (9.7) | 2 (6.4) | 11% |
| 20–29 years (N = 1,209) | 187 (65) | 68 (36.4) | 66 (35.3) | 26 (13.9) | 27 (14.4) | 15.5% |
| 30–39 years (N = 504) | 61 (21.2) | 28 (45.9) | 16 (26.2) | 7 (11.5) | 10 (16.4) | 12.1% |
| 40–49 years (N = 101) | 6 (2.1) | 5 (83.3) | 1 (16.7) | 0 | 0 | 5.9% |
| 50–59 years (N = 12) | 1 (0.3) | 1 (100) | 0 | 0 | 0 | 8.3% |
| Unknown age (N = 3) | 1 (0.3) | 0 | 1 (100) | 0 | 0 | 33.3% |
| Total | | 110 | 103 | 36 | 39 | |

**Table 4.** Suicide deaths by method

| | No. (%) suicides (N = 288) | No. (%) suicides | | | |
|---|---|---|---|---|---|
| | | Non-maternal | Antepartum | Puerperium | Postpartum |
| Poisoning | 82 (28.5) | 29 (35.4) | 31 (37.8) | 9 (11) | 13 (15.8) |
| Hanging | 67 (23.3) | 28 (41.8) | 24 (35.8) | 8 (12) | 7 (10.4) |
| Overdose | 47 (16.3) | 16 (34) | 20 (42.6) | 3 (6.4) | 8 (17) |
| Unknown | 43 (14.9) | 20 (46.5) | 5 (11.6) | 10 (23.3) | 8 (18.6) |
| Self-stabbing/cutting | 19 (6.6) | 3 (15.8) | 10 (52.6) | 3 (15.8) | 3 (15.8) |
| Self-immolation | 11 (3.8) | 5 (45.5) | 5 (45.5) | 1 (9) | 0 |
| Drowning | 10 (3.5) | 5 (50) | 4 (40) | 1 (10) | 0 |
| Jump from height | 5 (1.7) | 1 (20) | 3 (60) | 1 (20) | 0 |
| Discontinued HIV ART | 2 (0.7) | 1 (50) | 1 (50) | 0 | 0 |
| Railway track strike | 2 (0.7) | 2 (100) | 0 | 0 | 0 |
| Total | | 110 | 103 | 36 | 39 |

## Determinants of Suicide

Text data from death reports highlighted the circumstances surrounding FSW suicides. Across 77 suicide reports, peer informants identified several determinants of suicide: psychological distress, economic hardship impeding their ability to provide for their children, HIV diagnosis, and unplanned pregnancy. A diagnosis of HIV was identified as a reason for suicide (n = 24, 31.2%), the majority of which were in Kenya (n = 16, 66.7%). The diagnosis was described by multiple respondents as a "death sentence," citing bleak socioeconomic prospects. Among suicides associated with a new HIV diagnosis, cases were divided into non-maternal (n = 13,

54.2%) and maternal (*n* = 11, 45.8%). The majority of maternal suicides occurred while pregnant (*n* = 8, 72.7%) and the remainder occurred after childbirth in the puerperium and postpartum periods (*n* = 3, 27.3%). The two cases of HIV-related puerperium suicide were associated with the infant also testing positive, and one of these two mothers committed filicide-suicide.

In addition to reports of not being able to cope or having no support, participants described not being able to care for children and incidents of abuse or violence by clients and intimate partners (e.g., boyfriends, husbands) as contributors to suicide. A new unplanned pregnancy as a motivator for suicide was only reported in the DRC. Peer informants described the rejection of a child by the father as a rejection of both the mother and child that increased the risk of the mother resorting to suicide. Finally, exposure as a sex worker and the ensuing family rejection, particularly by the maternal figure, was identified as another determinant of suicide among FSWs in the DRC and Kenya.

### Children left without mothers

Five hundred four children lost their mothers to suicide (Table 5). Most children lost their mothers to non-maternal suicides (*n* = 230, 45.6%), followed by maternal suicides during pregnancy (*n* = 137, 27.2%) and postpartum (*n* = 79, 15.7%) and puerperium (*n* = 58, 11.5%) periods. For maternal suicides following delivery, the average age of the youngest child was 3 months old and ranged from 1 week to 12 months of age. Each suicide resulted in 0.857 to 2.175 children losing their mother, ranging from the lowest child-to-suicide ratio in Indonesia to the highest in the DRC.

### Interrater reliability

Cohen's kappa statistic indicated near-perfect agreement ($\kappa$: 0.94, 95% CI 0.91–0.97) between the two independent coders of pregnancy status and method of suicide. Classification discrepancies occurred exclusively with the "overdose" and "poisoning" categories, and seven overdose deaths were reclassified as poisonings following discussion. Three non-maternal vs. maternal discrepancies were detected and determined to be data transcription errors rather than misclassifications.

### Discussion

Death by suicide is increasingly recognized as a leading cause of maternal mortality globally, although research in this important

area of investigation is limited, particularly for stigmatized populations (Lindahl et al., 2005; Chin et al., 2022). This study presents the realities of suicide among a stigmatized subgroup of women in LMICs, focusing on maternal suicide. Our findings reveal a pattern of death by suicide, especially during the perinatal period, highlighting the need for targeted interventions and social support within this vulnerable population. This is also the first study to report on filicide-suicide among FSWs in LMICs.

The high prevalence of suicide we found among FSWs, particularly during pregnancy, underscores the acute vulnerability of pregnant FSWs. This finding challenges the conventional understanding that the postpartum period is the most critical phase for suicide risk. Instead, our study indicates that the entire perinatal continuum warrants attention for suicide prevention and that unique conditions of risk in this period are faced by sex workers. The elevated risk during pregnancy could be attributed to the compounded stressors of sex work, social stigma, and the lack of support systems, accentuating the need for comprehensive care approaches (Chin et al., 2022).

Compared to other perinatal populations, suicide rates in our study are alarmingly high. This difference is indicative of the magnitude of challenges faced by FSWs, such as increased exposure to violence, stigma, and mental health issues such as depression and post-traumatic stress disorder (Beattie et al., 2020; Millan-Alanis et al., 2021; Argento et al., 2021), as compared to the general population. In the current study, violence by clients and intimate partners, rejection by family and intimate partners, lack of child-rearing support, and a new HIV diagnosis were found to be important determinants of FSW suicide. This finding reinforces the notion that general suicide prevention strategies may not be sufficient for FSWs, who require more tailored approaches that consider their specific situations.

The significant association of suicide with an HIV diagnosis in this population is a critical finding. The perception of HIV as a "death sentence," coupled with socioeconomic hardships and sex work discrimination, creates a sense of hopelessness that may drive FSWs toward suicide. This finding emphasizes the importance of sustained efforts to combat the stigma surrounding HIV and the need to expand non-discriminatory care to FSWs, which may include access to HIV prevention (e.g., pre-exposure prophylaxis), HIV testing with counseling, uninterrupted antiretroviral therapy, and longitudinal HIV treatment programs with home-based medication, case management, and integrated mental health services.

Additionally, this study reported 19 filicide deaths of children as young as 2 months old. Despite scant literature on filicide-suicide,

**Table 5.** Children left without mothers

| Country | No. children who lost mother | No. suicides | No. (%) children impacted | | | | Child-to-suicide ratio |
|---|---|---|---|---|---|---|---|
| | | | Non-maternal | Antepartum | Puerperium | Postpartum | |
| Brazil | 1 | 1 | 1 (100) | 0 | 0 | 0 | 1.0 |
| DRC | 174 | 80 | 88 (50.5) | 64 (36.8) | 9 (5.2) | 13 (7.5) | 2.175 |
| India | 41 | 26 | 28 (68.3) | 8 (19.5) | 5 (12.2) | 0 | 1.577 |
| Indonesia | 6 | 7 | 2 (33.3) | 2 (33.3) | 0 | 2 (33.3) | 0.857 |
| Kenya | 135 | 90 | 67 (49.7) | 24 (17.8) | 18 (13.3) | 26 (19.2) | 1.5 |
| Nigeria | 125 | 72 | 36 (28.8) | 32 (25.6) | 22 (17.6) | 35 (28) | 1.736 |
| South Africa | 22 | 12 | 8 (36.4) | 7 (31.8) | 4 (18.2) | 3 (13.6) | 1.833 |
| Total | 504 | 288 | 230 | 137 | 58 | 79 | |

one previous study in the US reported "relationship issues" as a reason for female filicide-suicide (Murfree et al., 2022). While "relationship issues" was a common determinant of suicide cited among participants in the current study, a new diagnosis of HIV was the most reported determinant in the case of filicide-suicides. These acts of violence against children may be preventable if the determinants of filicide-suicide were better understood and properly addressed.

Consistent with suicide estimates for women in India (WHO, 2021a), this study found that suicides as a proportion of the total number of country deaths reported was greatest in India (19.7%) followed by Kenya (17.0%) and Nigeria (13.7%). Because India is the only country where neither the selling nor soliciting of sex is criminalized if done privately (GNSWP, 2021), this finding seems counterintuitive when considering the potential role of sex work criminalization and discrimination in suicide decisions. Interestingly, of the eight study countries, India, Kenya, and Nigeria have or had until recently laws against suicide attempts (UNGMH, 2021; Lew et al., 2022), raising the possibility that FSWs in these countries are not only deterred from accessing mental health services but may also tend to employ more definitive methods of suicide to improve odds of completion, hence the greater proportion of deaths attributed to suicide.

Consistent with global estimates (WHO, 2021a), the current study found that death by suicide was most prevalent among the 20 to 29-year-old age group, followed by 30- to 39-year and 15 to 19-year-old sex workers. It is perhaps unsurprising to find that the majority of maternal suicides were reported in the DRC, Kenya, and Nigeria, given that Sub-Saharan Africa alone accounted for approximately 70% of global maternal deaths in 2020 (WHO, 2023).

The impact of maternal suicides on children is another critical aspect of our study. The loss of a mother to suicide leaves children in precarious situations, often exacerbating their vulnerability to social and economic adversities. This finding underscores the necessity of developing child welfare programs alongside interventions for FSWs. Finally, efforts are urgently needed to strengthen both the community support available to FSW mothers and their children during the perinatal period, and the legal frameworks available to protect FSW from violence by clients and intimate partners.

### Strengths and limitations

The methodological strengths and limitations of the study are worth highlighting. Discussions related to mental illness, depression, and suicide could exacerbate the stigma experienced by an already profoundly marginalized population. Mental health stigma can result in significant cultural, social, and gender-related barriers to discussions of suicide. Although social desirability bias could potentially hamper information-sharing, particularly related to reports of filicide-suicide in their peer community, the direct involvement of trusted staff from known local SWO and NGO partners may have mitigated the impact of social desirability bias. As stigma likely affects the ability to elicit these outcomes, our measures would then likely be a conservative measure of this outcome and thus, not affect the significance of our findings of high percentages of maternal mortality attributable to suicide.

Similarly, while it is difficult to speculate on the effect of gender/race discordance between the lead investigator and study participants when reporting sensitive information, it is possible that the gender/race concordance of familiar local partner staff assisting in or conducting the group sessions may have, at least in part, countered any social desirability bias leading to minimization or exaggeration of information.

Due to the need for local partners who have established trust with this hard-to-reach, marginalized population of women, convenience sampling of FSW communities limits the ability to survey a sample of FSW communities that are more representative of the overall FSW population in any given country. For example, based on the geographic location of local partners, the study population was predominantly urban-based and less representative of rural FSW communities. Additionally, related to the recruitment of individual FSW participants, both convenience and snowball sampling represent non-probability sampling methods that may introduce selection bias into the sample, calling into question the generalizability of findings.

The use of the CKA method to elicit reports of suicide and explore the details surrounding these deaths allowed for an intimate gathering of peers where the collective memory of the group could be harnessed. On the other hand, in some cases, time constraints (60-minute sessions) made it necessary to terminate information-gathering before fully exhausting the memory of participants. Additionally, while the CKA is a low-cost, low-resource, validated method for collecting accurate and reliable information on the deaths of women and children within a community (Paul et al., 2018; Gupta et al., 2021), the current study design does not include a measure of internal validity. Household survey verification of deaths is challenging in FSW populations as this process would require that households of origin be: (1) aware of their family member's nature of work, deceased status, and maternal status, (2) knowledgeable of the details and circumstances surrounding their death, and (3) willing to openly discuss a potentially stigmatizing and emotionally disturbing cause of death in the family with a stranger. Research with highly marginalized groups requires a willingness to depart from widely accepted traditional methodologies and develop or adapt new equally effective methodologies in order to mitigate the risk of harm. Research conducted without regard for the potential harmful impact of the study design should not be undertaken at the expense of participants' and other stakeholders' psychological safety and well-being.

Finally, the inability to collect data in more than an average of three cities per country not only limits the extent to which these data are representative of deaths across all FSW communities in a given country, but without accurate size estimates of the FSW population, country mortality rates cannot be estimated. Therefore, while useful for identifying local trends on causes of death, these data cannot be used to estimate mortality rates or draw meaningful comparisons across countries. Furthermore, while this study does not account for non-fatal suicide attempts, which may provide a more comprehensive view of mental health, the percentage of deaths by suicide represents a generally more reliably recorded measure of outcomes than suicide attempts and is the outcome that public health interventions target for reduction.

### Conclusion

This represents the first multicountry study to investigate death by suicide among a particularly marginalized subgroup of women in LMICs. Results provide insight into the methods of suicide used among FSWs, and the unique circumstances potentially functioning as key determinants or contributors of suicide in this group. Although sex workers experienced physical and psychological

trauma, additional stressors such as economic hardship, new HIV diagnosis, unplanned pregnancy, rejection of a child by the father, and family rejection were associated with suicide and filicide-suicide. With increased attention to the health and socioeconomic needs of mothers who are sex workers and their children, the risk of suicides and filicide-suicides may be modifiable.

Our findings suggest a multifaceted approach may be necessary to address the issue of pregnant FSW suicide and may include suicide prevention strategies and mental health treatment services. More research is needed to determine the full scope of interventions needed to effectively address this outcome among FSWs who are mothers or are pregnant.

**Open peer review.** To view the open peer review materials for this article, please visit http://doi.org/10.1017/gmh.2024.74.

**Data availability.** Due to ethical considerations, controlled and secure data access and usage is necessary for subject protection. The lead investigator (BW) will consider granting access permission to the de-identified aggregate data used for this analysis based on the following usage criteria: (a) for the purpose of partnering on research on FSWs or (b) for the provision of services to female sex workers and their children by government and non-governmental organizations.

**Acknowledgements.** We would like to thank all the women who participated in the study and our local partners, including: Swasti (India); Bar Hostess Empowerment and Support Programme (BHESP; Kenya); SWOP Ambassadors (Kenya); Coast Sex Workers Alliance (COSWA; Kenya); Kisumu Sex Workers Alliance (KISWA; Kenya); Partners For Health & Development in Africa (PHDA; Kenya); Nigeria Sex Workers Association (NSWA; Nigeria); Royal Women Health and Rights Initiative (Nigeria); Initiative for Young Women's Health and Development (Nigeria); Action Humanitaire pour la Sante et le Developpement Communautaire (AHUSA- DEC; DRC); Cadre de Recuperation et d'Encadrement pour l'Epanouissement Integral des Jeunes (CREEIJ; DRC); and the Association pour le Soutien, l'Education, la Promotion de la Vie et des Initiatives Communautaires (ASEPROVIC; DRC), and Acção deSolidariedade e Saúde Comunitaria (ASSC – ONG; Angola).

**Author contribution.** WMK contributed to the conceptualization of the analysis, data coding and analysis, data visualization, and interpretation of results, and led the writing and revising of the manuscript. BW contributed to the conceptualization of the study, study design, procurement of funding, study administration, training and supervision of local staff, data collection, entry, and analysis, and interpretation of results. SW contributed to data coding, text data analysis, interpretation of results, and writing relevant sections. EP contributed to data curation and data analysis. IMB contributed to reviewing and editing the manuscript for critical content. All authors had full access to the study data, reviewed and approved the final version to be published, and accept responsibility for all aspects of the work.

**Financial support.** This work was supported, in whole or part, by the New Venture Fund and the Bill & Melinda Gates Foundation [INV-049925]. Under the grant conditions of the Foundation, a Creative Commons Attribution 4.0 Generic License has already been assigned to the Author Acceptance Manuscript version that might arise from this submission.

**Competing interest.** The authors have no Competing interest to declare.

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
