## [Editor Report]

There is agreement that this is valuable research, addressing an understudied and highly vulnerable population in LMIC. This manuscript can help to better understand the problem and the development of protective interventions for this group of sex workers.

The reviewers made recommendations to improve the different chapters of the manuscript, as well as some specific observations that the authors should correct.

Please consider the different comments and submit a new version of your manuscript.

---

## [Editor Report]

The current version of the manuscript has incorporated most of the recommendations that were made and the text is much clearer.

The work is almost accepted, but the authors are required to incorporate some adjustments within the text, which are noted below (this a proposal on the tone, vocabulary, and grammar to maintain the formality and passive language typically found in protocols):

Page 8 of 56 line 6. Change ¨Between January and October¨ to ¨From January to October¨

Page 8 of 56 line 10. Change the number ¨1,280¨ to ¨One thousand two hundred eighty¨

Page 8 of 56 line 12. Change the number 2, 112 to ¨Two thousand one hundred twelve¨

Page 8 of 56 line 14. Re-write ¨Of these maternal suicide, 57.9% occurred during pregnancy (antepartum)¨

Page 8 of 56 line 16. Re-write ¨The highest proportion of suicides occurred in Nigeria, Kenya and DR in sub-Saharan Africa.¨

Page 10 of 56 line 3. No need to write again ¨low- and middle-income countries (LMICs)¨, just write ¨LMICs¨without the parentheses.

Page 12 of 56 line 1. Change ¨Among a cohort¨ to ¨In a cohort¨

Page 12 of 56 line 25. Change ¨the methods of suicide used¨ to ¨the suicide methods¨

Page 13 of 56 line 1. Change ¨between January 16 and October 1¨ to ¨from January 16 to October 1¨ and delete the adjective word ¨large¨.

Page 13 of 56 line 3. I don’t quite understand the meaning/purpose of the line ¨Study methods were described in detail elsewhere and briefly reviewed here (Willis et al., 2022)¨, seems out of nowhere/place.

Page 13 of 56 subtitle ¨Study Participants and Recruitment¨ delete the word ¨Study¨

Page 13 of 56 line 29. Change ¨mother to at least¨ to ¨mother of at least¨

Page 14 of 56 line 11. Delete ¨as identified by local partners¨ is not necessary anymore.

Page 14 of 56 line 18. Delete ¨During the study period indicated¨and start directly with ¨Information-gathering group¨.

Page 14 of 56 line 19. Delete the word ¨study¨ of the phrase¨ eight study countries¨.

Page 14 of 56 line 22. Change ¨Where language translation¨ to ¨When language translation¨

Page 14 of 56 line 23. Delete ¨trained and¨

Page 14 of 56 line 35. Change ¨between 2014 and the session date¨ to ¨from 2014 to the session date¨

Page 14 of 56 lines 37 and 38. ¨, including her name; date of death; age at the time of death; pregnancy status; gestation age at the time of death, if pregnant; city of death; cause of death;¨ please change all those semicolons to simple commas.

Page 14 line 39. Chance ¨ In cases of suicide deaths, participants were asked about the method and location of suicide¨ to ¨In cases of suicide deaths the method and location of suicide were collected¨

Page 15 of 56 line 4. Delete ¨Following each group session¨ and start with ¨The lead investigator…¨

Page 15 of 56 line 5. Change ¨If any two deaths matched on two reported details¨ to ¨If any deaths matched on reported details¨

Page 15 of 56 line 9. Delete ¨The current analysis contains the subset of deaths due to suicide¨ and start with ¨Suicides were organized…¨

Page 15 of 56 line 10. Delete ¨if maternal¨ is not necessary.

Page 15 of 56 line 21. Delete ¨Based on details of the deaths¨and start with ¨Methods of suicide…¨ and also delete the word ¨study¨ before investigators, is not necessary.

Page 15 of 56 line 23. Delete ¨For ease of comparison, each rater created a¨ and start with ¨Data table….¨and after ¨perinatal period¨ delete the period to add ¨were created and¨ finally delete ¨the data tables were¨ the result must be like this: ¨Data table displaying the number of deaths by suicide method and by non-maternal vs. perinatal period were created and shared with all team members…¨

Page 16 of 56 line 13. Start the results without ¨A total of¨ and directly write ¨One hundred sixty five information-gathering…¨ also, change the period for a comma after sites and the ¨in the total¨, and merge with the rest, write the number, the result must be: ¨One hundred sixty five information-gathering group sessions were convened across study sites, one thousand two hundred eighty FSWs participated…¨

Page 16 of 56 line 14. Change ¨they reported a total of 2,112 FSW deaths, representing…¨ to ¨two thousand one hundred twelve FSW deaths were reported, representing…¨

Page 16 of 56 line 21. Change ¨while pregnant¨ to ¨during the pregnancy¨

Page 16 of 56 line 22. Write ¨Two months postpartum¨ instead of ¨2 months…¨

Page 16 of 56 line 23. Write ¨Two to twelve months¨ instead of ¨2 to 12-months¨

Page 17 of 56 line 3. Write ¨ The distribution of suicide deaths by age group is provided in Table 3.¨

Page 17 of 56 line 5. Delete the second among, it should be: ¨…occurred among FSWs aged 20-29 years, whom suicide….¨

Page 17 of 56 line 7. Delete ¨olds¨ in ¨ 30-39-year-olds and 15-19-year-olds¨ it should be: ¨ 30-39-year and 15-19-year¨

Page 17 of 56 line 12. Delete ¨use¨, it should be: ¨Other methods of suicide included self-inflicted…¨

Page 17 of 56 line 16. Change ¨Up to three children were killed per episode of filicide-suicides. When known, reasons for….¨ to ¨Up to three children died in filicide-suicide episode. Reasons for filicide-suicide included family rejection…¨

Page 17 of 56 line 18 and 19. The percentage is missing in Poisoning (n=9, %?), hanging (n=2, %?) and drowning (n=1, %?), please add those since the other ones included this.

Page 17 of 56 line 22. Delete the word ¨salient¨ in the phrase ¨several salient determinants¨

Page 18 of 56 lines 1-6. Edit those numbers according to the format previously used (n=x, %), my suggestion would be: ¨A diagnosis of HIV was identified as a reason for suicide (n=24, 31.2%), the majority of which were in Kenya (n=16, 66.7%). The diagnosis was described by multiple respondents as a “death sentence” citing bleak socioeconomic prospects. Among suicides associated with a new HIV diagnosis, cases were divided into non-maternal (n=13, 54.2%) and maternal while pregnant (n=8, 33.3%).

Page 18 of 56 line 5-6. After the phrase ¨and the remainder occurred after childbirth in the puerperium and postpartum periods¨ please add their corresponding number and percentage as previously said (n=x, %)

Page 18 of 56 line 17. Delete ¨Across the study¨ and start with the written number ¨Five hundred four children lost…¨

Page 18 of 56 line 18. Delete ¨ followed in descending order by¨ it doesn’t seem necessary.

Page 18 of 56 line 23. The cohen kappa written wasn’t added to the methodology, please add it, it seems out of nowhere.

Page 19 of 56 line 3. Change ¨major source¨ to ¨leading cause¨

Page 19 of 56 line 4. Change ¨globally though the work in this important area of investigation is limited..¨ to ¨globally, although research in this important area of investigation is limited..¨

Page 19 of 56 line 6. Delete the phrase ¨an important look into¨, is not necessary.

Page 19 of 56 line 6-7. Change ¨among a highly stigmatized subgroup of women in LMICs, with a particular focus on maternal suicide¨ to ¨among a stigmatized subgroup of women in LMICs, focusing on maternal suicide.¨

Page 19 of 56 line 8. Change ¨highlighting a need for targeted..¨ to ¨highlighting the need for targeted..¨

Page 19 of 56 line 10. Delete the word ¨context¨

Page 19 of 56 line 16. Change ¨societal stigma¨ to ¨social stigma¨

Page 19 of 56 line 19. Change ¨When compared to other perinatal populations, the suicide rates¨ to ¨Compared to other perinatal populations, suicide rates…¨

Page 19 of 56 line 21. Change ¨ issues like depression¨ to ¨issues such as depression¨

Page 19 of 56 line 26. Change ¨may not suffice for FSW¨ to ¨may not be sufficient¨

Page 20 of 56 lines 12-13. Delete ¨where a determinant was reported¨ seems redundant

Page 21 of 56 line 4-6. Change ¨ Finally, efforts to strengthen both the community supports available to FSW mothers and their children during the perinatal period, and the legal frameworks available to protect FSW from violence by clients and intimate partners are urgently needed¨ to ¨Finally, efforts are urgently needed to strengthen both the community…¨ are urgently needed should be at the beginning as I wrote.

Page 21 of 56 line 11. Change ¨While social desirability¨ to ¨Although social desirability¨

Page 21 of 56 line 14. Delete ¨at least in part¨ phrase

Page 22 of 56 line 5. Delete ¨and the participation of a few could encourage the participation of many¨ phrases, is not necessary.

Page 23 of 56 line 1. Change ¨The current study represents the first multi-country study…¨ to ¨This represents the first multi-country study…¨

Page 23 of 56 line 2. Delete ¨of this study¨ in the phrase ¨Results of this study provide¨, seems redundant.

Page 23 of 56 line 4. Change ¨While sex workers¨ to ¨Although sex workers¨

Table 1. Can be named just ¨Study Countries and Suicide Deaths¨ the summary of is not really necessary

Table 3. Is not relly necessary the division ¨No. suicides (% of suicides in age group)¨, it would be better without it.

Table 5. Can be named ¨Children Left Without Mothers¨ the summary of is not really necessary

My recommendations for all the tables: Use the rule (n=, %) on the title of each division, ex on the first table: No. sessions (n=165), No. session participants (n=1,280), No. suicides n=110 (38.2%), etc. The total is not really necessary with this and also you can exclude all the % used, since the table has a lot of these and doesn’t look great.

It would also be very valuable if the authors added a comment on the measures taken to provide psychosocial support after women gave their accounts of suicide and filicide-suicide in the group discussions, whether the local partners had the capacity to identify women experiencing distress, or whether the CKA method addresses this need.

---

## [Editor Report]

The latest version has been reviewed and there is agreement that the authors have introduced the suggestions within the manuscript. The topic it addresses is of great interest and there are few studies on it.

It is good work and deserves to be published.